# Fifty Years of the Fluid–Mosaic Model of Biomembrane Structure and Organization and Its Importance in Biomedicine with Particular Emphasis on Membrane Lipid Replacement

**DOI:** 10.3390/biomedicines10071711

**Published:** 2022-07-15

**Authors:** Garth L. Nicolson, Gonzalo Ferreira de Mattos

**Affiliations:** 1Department of Molecular Pathology, The Institute for Molecular Medicine, Huntington Beach, CA 92647, USA; 2Laboratory of Ion Channels, Biological Membranes and Cell Signaling, Department of Biophysics, Facultad de Medicina, Universidad de la República, Montevideo 11800, Uruguay; ferreiragon@gmail.com

**Keywords:** lipid interactions, membrane domains, extracellular matrix, lipid rafts, membrane fusion, membrane structure, membrane dynamics, cytoskeletal interactions, membrane vesicles, endosomes, exosomes, detoxification, chronic medical conditions

## Abstract

The Fluid–Mosaic Model has been the accepted general or basic model for biomembrane structure and organization for the last 50 years. In order to establish a basic model for biomembranes, some general principles had to be established, such as thermodynamic assumptions, various molecular interactions, component dynamics, macromolecular organization and other features. Previous researchers placed most membrane proteins on the exterior and interior surfaces of lipid bilayers to form trimolecular structures or as lipoprotein units arranged as modular sheets. Such membrane models were structurally and thermodynamically unsound and did not allow independent lipid and protein lateral movements. The Fluid–Mosaic Membrane Model was the only model that accounted for these and other characteristics, such as membrane asymmetry, variable lateral movements of membrane components, cis- and transmembrane linkages and dynamic associations of membrane components into multimolecular complexes. The original version of the Fluid–Mosaic Membrane Model was never proposed as the ultimate molecular description of all biomembranes, but it did provide a basic framework for nanometer-scale biomembrane organization and dynamics. Because this model was based on available 1960s-era data, it could not explain all of the properties of various biomembranes discovered in subsequent years. However, the fundamental organizational and dynamic aspects of this model remain relevant to this day. After the first generation of this model was published, additional data on various structures associated with membranes were included, resulting in the addition of membrane-associated cytoskeletal, extracellular matrix and other structures, specialized lipid–lipid and lipid–protein domains, and other configurations that can affect membrane dynamics. The presence of such specialized membrane domains has significantly reduced the extent of the fluid lipid membrane matrix as first proposed, and biomembranes are now considered to be less fluid and more mosaic with some fluid areas, rather than a fluid matrix with predominantly mobile components. However, the fluid–lipid matrix regions remain very important in biomembranes, especially those involved in the binding and release of membrane lipid vesicles and the uptake of various nutrients. Membrane phospholipids can associate spontaneously to form lipid structures and vesicles that can fuse with various cellular membranes to transport lipids and other nutrients into cells and organelles and expel damaged lipids and toxic hydrophobic molecules from cells and tissues. This process and the clinical use of membrane phospholipid supplements has important implications for chronic illnesses and the support of healthy mitochondria, plasma membranes and other cellular membrane structures.

## 1. Introduction: Barriers, Cellular Compartments and Biomembrane Structure

When exogenous or extracellular molecules, including water, ions, nutrients, sugars, proteins, glycoproteins, lipids, lipoproteins and other components, such as extracellular structures, stroma, extracellular matrix, lipid vesicles, viruses, microorganisms, and other cells approach a cell, they first encounter cell membranes or cell membrane-associated structures [1,2]. Cell membranes and their associated structures are the most important barriers to cell entry and exit of molecules, ions, and other structures, allowing a unique intracellular microenvironment [2,3,4]. The interactions of extracellular molecules and stromal structures and cell membranes are important in maintaining the exclusion of extracellular molecules, and they are also critical in segregating membrane molecules, regulating cell polarity, modulating the exchange of molecules, initiating cellular signaling, and moderating the responses to and maintenance of many normal cellular processes [3,4,5].

Although cell membranes provide cells with barriers and compartmentalization, they also support cellular and tissue continuity. The barrier functions can also be quite selective and adjustable, and thus cells are capable of selectively transporting particular nutrients and other substances into cells and eventually into various cellular organelles, as well as transmitting certain molecular signals into and out of cells. These molecular signals and effectors can be various secreted ions and molecules, lipid-associated structures, lipid vesicles, such as exosomes, and other substances that can find their way to adjacent cells, tissues and distant organs, and in the process initiate changes in cell and tissue microenvironments. Inside cells, various intracellular membranes are responsible for the segregation of enzymatic processes, the biosynthesis and transport of various molecules, and generally the separation of basic cellular functions, such as energy production, replication, secretion and other cellular activities [3,4,5,6].

Although each biomembrane is unique in its detailed structure, composition, dynamics and function, there are some general structural and organizational principles that should apply to all cellular membranes and be present in any biomembrane model.

**Biomembrane General Principle No. 1:** 
*The importance of noncovalent forces in establishing and maintaining nanoscale biomembrane structure cannot be underestimated.*


**Biomembrane General Principle No. 2:** 
*Lipid bilayers are essential in providing a basic matrix and continuity for biomembranes.*


The first two general principles necessary for any biomembrane model should cover the most basic aspects of cellular membranes. Cell membranes are macromolecular structures that are present in aqueous solutions and whose components are largely held together by noncovalent forces. At their most basic organizational level they can be thought of as dynamic barrier matrix structures made up of amphipathic lipid and protein components that spontaneously associate into largely noncovalently bound macro-structures that exclude water interactions on their hydrophobic surfaces. In contrast, the hydrophilic portions of their structures interact with the aqueous environment and other hydrophilic and ionic molecules [3,5,6,7,8]. This concept was implied by the experiments of Langmuir, who studied the formation of oil layers on aqueous surfaces [7]. Using this methodological approach, it was estimated that red blood cells are surrounded by two layers of membrane lipids [9]. This was also consistent with Fricke’s findings from cell membrane capacitance experiments that estimated that cell membranes should be approximately 4 nm thick [10]. The historical representations that cell membranes are basically composed of a phospholipid bilayer matrix plus some membrane proteins has been reviewed elsewhere [11]. In one notion of how cell membranes are organized, it was proposed that cellular membranes are basically phospholipid bilayers that interact with flattened or beta-sheet-structured proteins via the hydrophilic head groups of membrane phospholipids and certain amino acids [12]. Visualization of this structure, primarily by transmission electron microscopy of erythrocytes and other cells fixed and stained with heavy metals and embedded in polymeric resins and transversely thin-sectioned, revealed what appeared in cross section to be tri-molecular layers of membrane components. This visual representation was promoted as support for the basic organization of cell membranes as a trimolecular, layered structure composed of protein–lipid–protein units (the Unit Membrane) [13]. A competing membrane model was subsequently proposed that was based on a monolayer of repeating subunits of lipoproteins without a matrix composed of a phospholipid bilayer [14]. None of the ‘sandwich’ or ‘lipoprotein subunit’ models of cell membrane structure proved to be correct [8].

**Biomembrane General Principle No. 3:** 
*Membrane components attempt to maximize compatible molecular interactions in order to approach the lowest free energy state.*


Any model for biomembranes should consider the most important forces that hold membranes together and resist the natural tendency to disorganize over time. The concept of exploiting noncovalent forces that match the hydrophobic portions of each integral membrane component and minimize nonmatching interactions drive the spontaneous assembly of membrane lipids and proteins [15,16]. This implies that biomembranes are composed of a lipid matrix composed of amphipathic phospholipids that self-assemble to form a lipid bilayer due to the free energy provided by the hydrophobic effect and van der Waals forces [15,16]. Into this lipid bilayer matrix, integral membrane proteins are thought to assemble and interact with membrane lipids mainly via hydrophobic forces and much less by hydrophilic forces between lipid head groups and the membrane proteins’ hydrophilic amino acids [3,8,15,16,17,18]. The physical state of membrane phospholipids is important in this process because the insertion of integral membrane proteins into a lipid bilayer matrix may be limited to regions of membrane where the lipid matrix allows protein penetration and intercalation into the lipid bilayer. In the process, membrane protein–lipid hydrophobic interactions must be thermodynamically favorable. Thus, in the membrane regions where phospholipid and protein molecular sorting can occur, the hydrophobic and van der Waals forces can be maximized, and the lowest free energy state can be approached [7,8,16]. Thus, the lateral, independent movements of membrane components are possible in a fluid matrix [3,8,15,16,18]. (The different interactions of lipids with other lipids to form various domains of differing lipid compositions will be discussed in a subsequent section.) The most stable membrane structure is one that maximizes hydrophobic interactions, stabilizes ionic interactions and couples different ionic charges in an attempt to approach the lowest free energy state [8,16].

**Biomembrane General Principle No. 4:** 
*Membrane proteins are a large and diverse group of molecules that can be placed into different classes depending on their interactions with membrane lipids and their abilities to intercalate into the membrane lipid bilayer matrix.*


In this general principle of biomembrane structure, we consider the protein components and how they are incorporated into the overall structure. There are large numbers of unique proteins that are present in various cellular membranes [15,19]. These membrane proteins can be assigned to different categories or classes based on their amino acid sequences, functions and properties. Operationally, in the Fluid–Mosaic Membrane model, they were consigned to three simple classes: integral, peripheral [8,15], and (added later) membrane-associated proteins [17,18]. The classic integral (or intrinsic) membrane proteins were depicted as globular, amphipathic proteins that were intercalated into lipid bilayers and stabilized mainly by hydrophobic forces (Figure 1) [8]. In this model, the integral membrane proteins were thought to penetrate into the membrane lipid bilayer matrix to various degrees, from completely spanning the membrane to barely infiltrating into the lipid bilayer [8,15]. These membrane proteins and glycoproteins have extensive alpha helical regions and lipid-interacting structures at their surfaces, and they represent families of transport, adhesion, signaling and other molecules that that can be potentially modulated by their adjoining lipids. At the time, peripheral membrane proteins were proposed to be attached to membranes mainly by electrostatic or other forces [8]. These peripheral membrane proteins were purported to be removeable from membranes without destroying basic membrane structure and continuity [8]. They were subsequently found to serve as important components in providing membrane stability, deformation, curvature, scaffolding and other characteristics, such as attachment points for enzymes and signaling complexes [17,18,19]. A few years after the publication of the Singer–Nicolson model [8], the other category was added: membrane-associated proteins [17]. Membrane-associated proteins can be globular in structure, but generally they are not amphipathic and not associated with the hydrophobic membrane lipid matrix, nor are they thought to be bound by mainly electrostatic forces. These proteins can also be transiently associated with membranes via interactions with integral membrane proteins or linked to lipid molecules instead of being intercalated to various degrees into the membrane lipid matrix and stabilized by hydrophobic forces [17,18]. The membrane-associated proteins were alleged to dynamically and intermittently provide connections between cell membranes and other intracellular components at the inner membrane surface and extracellular and stromal components at the outer membrane surface. Examples of membrane-associated components at the inner membrane surface could be intracellular proteins, enzymes, protein complexes and cytoskeletal elements or, at the outer surface, extracellular stromal and matrix elements [17,18]. Membrane-associated proteins are now thought to assist in maintaining membrane shape, structural integrity and dynamics as well as to provide linkages to other intracellular and extracellular proteins and glycoproteins [3,5,17,18,19]. 

Biomembranes in general are dynamic structures that can be disturbed, distorted, deformed, compressed or expanded by different forces [3,16,17,18]. Indeed, certain peripheral membrane proteins can bind to and cause membrane deformability by binding to biomembranes, in the process causing membrane curvature as a result of flexing and bending of membranes to fit the structures of these peripheral proteins [3,16,17,18,20,21]. In contrast, membrane-associated proteins are thought to act indirectly on membranes, usually through intermediate protein or lipid attachments. Membrane peripheral and membrane-associated proteins should be removable from membranes without disruption of the membrane’s basic structural integrity and continuity of its hydrophobic matrix [3,8,17,18]. Membrane-associated proteins can be present in the cell cytoplasm or outside cells and include cytoskeletal and signaling structures bound at the inner cell membrane surface or extracellular matrix and stromal components interacting at the outer extracellular membrane surface. These membrane-associated components can be quite dynamic and can stabilize or destabilize cellular membranes and their connections to other intracellular or extracellular structures [3,17,18]. Alternatively, they can be involved in stabilizing the dynamic properties of membranes and preventing membrane components from undergoing various lateral movements and consigning them to certain spaces, or they can participate in the directional movements or translocation of membrane complexes via energy-dependent processes [3,17,18]. Membrane-associated proteins are involved in maintaining or eliciting certain specific cellular processes, including: cell adhesion, stabilization, motility, growth, endocytosis, exocytosis and other important cellular functions [3,5,16,17,18].

## 2. The Fluid–Mosaic Model of Biomembrane Structure

Although various models of biomembrane structure have been presented in the literature over the last 50-years, the most accepted nanometer scale model of basic cell membrane structure remains the Fluid–Mosaic Membrane Model (Figure 1) [8]. This model has been criticized as an oversimplified and obsolete scheme for explaining the complex nature of cellular membranes and their hierarchical structural organization, as well as for its failure to account for some of the dynamic properties and domain organizations found in certain biomembranes [22,23,24]. In its defense, however, the Fluid–Mosaic Membrane Model was never intended to explain all aspects of membrane structure and dynamics, especially those discovered after 1972. Instead, it was generated to provide a basic minimal framework of cellular membrane organization and dynamics, not as an ultimate future description for all of the potential molecular arrangements and subtleties present in various cellular membranes.

**Biomembrane General Principle No. 5:** 
*When proposing a model for basic biomembrane structure and organization, it should be consistent with current data as well as with future discoveries.*


**Biomembrane General Principle No. 6:** 
*Biomembranes are asymmetric in their distribution of membrane proteins and glycoproteins, certain lipids and glycolipids and peripheral and membrane-associated proteins on inner and outer membrane surfaces.*


These two general principles incorporate some basic elements of biomembranes that are based on the published evidence that accumulated in the years before 1972. The Fluid–Mosaic Membrane Model was put forward to provide a simple framework for the basic organization of biomembranes, not as a detailed explanation of the structure, asymmetry, dynamics and functions of every biomembrane [8]. We believe that it has accomplished this goal for the last 50 years, but it obviously has required periodic updates [3,17,18]. The Fluid–Mosaic Membrane Model has demonstrated its usefulness, but only as a simplified, nanoscale representation of rudimentary biomembrane structure. In that respect, it still represents some of the more important elements of cell membrane architecture, including continuity, cooperativity and asymmetry of biomembranes, as well as some aspects of membrane dynamics [3,17,18,25]. 

The original Fluid–Mosaic Model accounted for membrane asymmetry by estimating the enormous amount of free energy required to flip amphipathic membrane components from one side of a lipid bilayer to the other side [8]. Every cell membrane studied thus far has been found to be asymmetric in terms of the display of membrane components on the interior and exterior sides of membranes, especially those that have attached carbohydrates [3,8,11,15,16,17,18,21,26]. This characteristic of biomembranes appears to be universal [18,26]. It makes perfect sense to have asymmetric structures that separate different cell compartments.

**Biomembrane General Principle No. 7:** 
*There is no universal membrane model that can explain or predict every newly discovered aspect of biomembrane structure, function or dynamics.*


We admit that it is virtually impossible to incorporate all of the data published over the years on biomembranes into a universal model of biomembrane structure. The goal here is to come up with a reasonable solution that fits best with the available data. With the limited data available 50 years ago, only a few of the potentially vast number of biomembrane characteristics could be discussed at the time in any detail [8]. Some membrane elements were briefly mentioned but not presented graphically in the original schematic of a biomembrane (Figure 1), such as membrane asymmetry, specialized lipid environments surrounding membrane proteins, and other characteristics [8]. Unfortunately, this has led to some quite literal interpretations of the Fluid–Mosaic Membrane Model and, we feel, undeserved criticism [22,23].

Although the essential elements of the Fluid–Mosaic Membrane Model have proven to be remarkably consistent with experimental findings at the nanoscale level over the last 50 years, it was inevitable that the original model could not explain all of the newly discovered properties of membranes, including recent findings on the fine structure and dynamics of protein and lipid components [3,18,22,23,24,25,26,27,28,29,30,31]. Importantly, the concept that membrane mosaic structures and membrane domains, such as lipid rafts and membrane protein complexes as well as cell membrane-associated structures, such as cytoskeletal elements and other structures, were essential in controlling membrane properties and directing the dynamics of certain cell membrane components. These were not features found in the original model, and many of these new findings were made decades after the publication of the original Fluid–Mosaic Model [3,18,22,23,24,25,26,27,28,29,30,31]. This has resulted in the suggestion that several membrane models are necessary to explain basic biomembrane structure and dynamics [22], or that there are no general membrane models that can adequately describe the structure and dynamics of biomembranes [23]. We understand the need to constantly update existing proposals. Moreover, we recognize the difficulty in presenting an accurate model for biomembrane structure and dynamics that takes into account all of the data accumulated since 1972 [3,18].

**Biomembrane General Principle No. 8:** 
*Biomembranes appear to be much more complex, compact and more mosaic than presented in the original*
*Fluid–Mosaic Membrane Model.*


There is a misconception over the first presentation of the Fluid–Mosaic Membrane Model (Figure 3 of Ref. [8], redrawn as Figure 1 in this contribution) that has persisted over the years since its publication in 1972. That is, the densities of protein components in biomembrane models have increased with time. Over the last 50 years. various updates of the Fluid–Mosaic Membrane Model have gradually refined the original model of biomembrane structure and dynamics into far more complex, much less homogeneous, and more densely packed (more mosaic) models, as emphasized by the authors [3,18] and Goñi [28]. Compared to the original biomembrane scheme, shown here as Figure 1 [8], newer membrane models depict biomembranes as more mosaic in nature with many fewer areas of fluid membrane lipid regions [3,18,22,23,24,25,26,27,28,29,30]. All of the newer proposals on biomembrane organization also contain additional information not shown in the original model (Figure 1), such as representations of lipid–lipid, protein–protein and lipid–protein associations into membrane domains of various sizes and surrounded by specific combinations of lipids as well as nano- and micro-sized complexes within specialized domains, and their segregation and regulation by transmembrane forces. Importantly, all of the newer biomembrane models now include membrane-associated structures on the cytoplasmic side and in the extracellular environment that are capable of immobilizing or alternatively mobilizing large portions of membrane. Most mammalian cells are located in tissues where polarity and cellular and extracellular and stromal interactions are important in segregating and maintaining tissue organization and cellular networks. In addition, newer information on transmembrane signaling complexes, membrane component interactions and dynamic changes in membrane organization, along with other additions had to be accommodated [18,19,20,21,22,23,24,25,26,27,28,29,30,31,32]. These additions over the years have made biomembrane organizational schemes much more complex and compact (more mosaic) than the original Fluid–Mosaic Model (for example, Figure 2). 

During the last decade, it has become fashionable to position most biomembrane lipids and proteins into less freely-mobile domains, such as lipid-rafts and lipid–protein complexes, or specialized membrane domains linked to cytoskeletal elements. Hence the mosaic nature of cellular membranes has now been accentuated over the fluid nature of membranes [3,18,28]. Although newer biomembrane models contain fewer fluid areas of freely mobile membrane lipids and proteins than presented in the original Singer–Nicolson model [3,18,22,24,28], the basic nanoscale organization first presented in the Fluid–Mosaic Membrane Model as diverse amphipathic proteins intercalated to various degrees into a lipid bilayer matrix has generally survived [3,18,22,23,24,25,26,27,28,29,30]. 

**Biomembrane General Principle No. 9:** 
*Biomembranes appear to have a more complex multicomponent hierarchical organization than originally envisioned*


This general principle of biomembranes signifies that for over 50 years new data on biomembrane structure and organization have necessitated some adjustments in the original Fluid–Mosaic Model. As described above, current biomembrane models are more crowded and complex (more mosaic) than presented in the original Singer–Nicolson proposal [3,18,28,29]. To add to this complexity, Kusumi and his colleagues have advanced the concept of a dynamic hierarchical cell membrane structural organization [24,25]. This has made a complicated description of cell membrane organization even more layered and complex in order to incorporate recent data on the presence of various macromolecular structures (superstructures) in some biomembranes. The various macromolecular structures appear to place restrictions on the distribution and mobility of some membrane components [24,25]. This will be described in more detail in a subsequent section of this review.

Any reasonable schematic of biomembrane organization should depict the nonrandom sorting and the various mobilities and distributions of different membrane components [27,29,31]. The spontaneous, dynamic sorting of membrane components into various membrane domains was thought to be based, at least initially, on hydrophobic and some hydrophilic interactions [3,15,18,32]. Such dynamic sorting avoids hydrophobic mismatches between various lipids and lipids and proteins, thus preventing unsustainable membrane distortions or areas of membrane weakness [32]. 

In the original Fluid–Mosaic Model, the presence of some oligomeric protein/glycoprotein structures in the membrane was first proposed (see Figure 1) [8]. Some early evidence (discussed in [8]) was the discovery of different cell surface antigen distributions—dispersed [33] or micro-clustered [34]—on the same cell type. That notion has now become more refined based on evidence gathered with new technologies developed to study the localization and dynamics of single molecules on cell surfaces at the nanometer scale [29,31,35]. For example, Garcia-Parajo and colleagues found that many, if not most, cell membrane proteins and glycoproteins exist in small mobile nanostructures or nanoclusters in the membrane [29]. Using Förester Resonance Energy Transfer (FRET) combined with single particle tracking and fluorescence microscopy, Ma et al. studied the associations of neighboring membrane proteins and their clustering events at high spatial and temporal resolutions [35]. By plotting the individual mobilities and clustering events on live cells, Pageon et al. found that certain receptors were already present in ‘nanoclusters’, and these dense receptor clusters were important in providing the greatest signaling efficiencies [36]. Membrane domain dynamics involve lipid–lipid and lipid–protein interactions as well as inner membrane surface protein scaffolding and the involvement of membrane-associated elements [37]. This will be discussed again in a following section.

Over time, the basic nanoscale organization of cell membrane models has evolved significantly from the original models of rather homogeneous-looking structures, such as the diagram shown in Figure 1 [8], into more heterogeneous models that are still dynamic yet contain mosaic structures that comprise specific domains of varying sizes, compositions and component mobilities. Some of these structures can form into specific membrane regulatory and mechanical platforms that are linked to various intra- and extracellular components (cytoskeleton structures, stromal components, extracellular matrix, etc.) [3,18,22,24,25]. 

## 3. Some Important Interactions of Proteins and Lipids in Biomembranes

In the original Fluid–Mosaic Membrane Model, membrane components were, in general, portrayed as primarily randomly distributed and unrestrained in their lateral movements [8]. However, as mentioned in Section 2, certain properties, such as the variable lateral mobilities of many membrane components, are now assumed to be part of the model [3,17,18]. Within a few years after the original model was presented, these concepts were incorporated into an updated Fluid–Mosaic Membrane Model [17]. 

**Biomembrane General Principle No. 10:** 
*Changes in the compositions of certain asymmetrically distributed membrane lipids can modify the physical characteristics of biomembranes.*


This general principle of biomembranes reflects the many studies on the properties of membrane phospholipid and other membrane lipids over the last 60 years. Biomembranes are known to contain hundreds of different types of lipids, most in minute concentrations [11]. We do not know the functional consequences of having very-low-abundance membrane lipids in cellular membranes, although the complete absence of specific lipids has known clinical consequences. Moreover, it is accepted that certain specific membrane phospholipids can form specialized zones around the hydrophobic surfaces of membrane proteins, possibly due to their stronger interactions with the hydrophobic surfaces of membrane integral proteins and glycoproteins, and to a lesser degree to hydrophilic interactions [32,38,39,40]. In addition, certain membrane phospholipids have been found to be asymmetrically present on the inner and outer leaflets of plasma membranes and also unevenly distributed in the membrane plane in certain lipid domains, and this could have important consequences on membrane curvature, flexibility and other properties [3,32,38,39,40,41]. In addition to membrane phospholipids, other lipids are also distributed nonrandomly in various cellular membranes. For example, cholesterol is intercalated into biomembrane bilayers and can modify the characteristics of membranes and change lateral lipid distributions and dynamics [32,38,39]. Cholesterol is often found to be enriched within specific membrane domains [38,41]. Its distribution is thought to be due, in part, to its affinity for both the fluid and solid phases of membrane phospholipids [38,41]. Cholesterol partitions into liquid-ordered and disordered lipid phases to roughly the same extent, but this partitioning can modify the properties of dissimilar membrane lipid phases [38,41]. Changes in the compositions of membrane phospholipids can also modulate certain physical properties of membranes [42]. For example, changing cholesterol content can modify membrane lateral elasticity, whereas ceramides and lysophospholipids are known to induce changes in membrane curvature [38,41,42].

**Biomembrane General Principle No. 11:** 
*Biomembrane lipids in specialized lipid–protein domains are essential in maintaining membrane structure and function.*


Specific membrane lipids, for example sphingolipids, are important in the formation of ordered membrane lipid mosaic domains or ‘lipid rafts’ [38,43,44,45,46,47]. With phosphatidylcholine, sphingomyelins constitute more than one-half of the cell membrane phospholipids and are the most important companions of cholesterol in lipid domains or lipid rafts [47,48]. Small, ordered membrane rafts/domains assemble by preferential associations of cholesterol and saturated lipids. These rafts/domains are generally surrounded by liquid-phase lipids, and thus they are able to undergo membrane lateral movements [46,47]. Lipid rafts/domains can also selectively recruit additional lipids and proteins into their structures [30,43,44,45,46,47]. Not all of the lipids within such mosaic domains are completely immobilized—they are still rotationally and laterally mobile to some degree and capable of slowly exchanging their lipids with bulk membrane lipids as well as with lipids in other membrane domains [30,46,47]. The overall sizes of lipid domains, such as lipid rafts, are usually less than 300 nm in diameter; most are within 10–200 nm in diameter, with some slightly larger [49,50,51]. However, they can undergo domain clustering induced by protein–protein and protein–lipid interactions, and the result is an increase in domain diameter to approximately micrometer size (>300 nm) [30,50]. 

Lipid rafts/domains usually contain some peripheral membrane proteins and lipid-linked proteins as well as some integral membrane proteins, and these mixed lipid–protein domains are not static [50,51]. Lipid rafts/domains undergo changes in lipid and/or protein compositions over time and can convert these membrane platforms into functional signal transduction domains. Eventually, the transmembrane-coupled rafts/domains can initiate various functions, such as immune receptor signaling, host–pathogen reactions, cell-death regulation and other cellular processes [43,45,46,50,51]. 

Membrane proteins can have profound effects on biomembranes and on lipids within lipid rafts/domains. They can deform membranes and cause reorganization of membrane lipids to form new membrane domains as well as regulate various membrane properties, such as charge density and diffusion rates [21,52]. When integral membrane proteins interact with membrane lipids in biomembranes, portions of their structures must directly interact with the acyl chains of membrane phospholipids or other hydrophobic regions of other molecules. This is accomplished by hydrophobic matching between the hydrophobic regions of proteins and lipids [32,38]. Hydrophobic matching between the hydrophobic core of the lipid bilayer and hydrophobic stretches of amino acids in integral membrane proteins results in stable hydrophobic interactions by the exclusion of water. If the hydrophobic portions of their structures are mismatched, elastic distortion of the lipid matrix around the integral membrane protein occurs [32,38]. This can induce protein conformational changes that can affect protein function and protein–protein and protein–lipid interactions. Membrane proteins can also aggregate to form super-domains in membranes. In addition, there are other physical forces, such as lateral pressure forces, lateral phase changes, membrane curvature, ionic interactions and other forces, that are important in regulating membrane structure, function and dynamics [52,53,54]. 

## 4. Membrane-Associated Cytoskeletal and Extracellular Matrix Interactions with Biomembranes

Negligible information was available on membrane interactions with intracellular cytoskeleton networks and extracellular matrix elements at the time of the original Singer–Nicolson publication [8]. Although such interactions were assumed to be important in the attachment of cells to substratum and stroma, at the time, the components involved in these interactions were not well-characterized [55]. That cell membrane-associated interactions could alter cell membrane macrostructure by restricting the dynamics or lateral movements of membrane proteins and glycoproteins and segregating them into membrane domains was virtually unknown at the time. In addition, important membrane properties, such as cytoskeleton–membrane linkages are now known to be involved in immobilizing membrane domains as well as moving domains with the assistance of energy-dependent cytoskeletal processes [3,17,18,27,28,29,30,31,56,57]. Indeed, cytoskeletal and extracellular elements are now known to be essential in maintaining cell polarity and tissue organization [56,57,58,59,60,61,62,63,64]. 

**Biomembrane General Principle No. 12:** 
*Biomembranes are not automatous structures—their components and domains are linked and integrated with various intracellular and extracellular structures.*


We will now consider some of the more complex properties of biomembranes that are related to their many functions. There are a number of tissue and cellular properties, such as cell orientation, cell adhesion, cell movement, cell stabilization, cell communication, cell differentiation and many other properties, that are driven or stabilized by membrane-associated cytoskeletal interactions with membrane domains [3,17,18,25,51,56,57,58,59,60,64,65]. Cell membrane receptor clustering, domain formation, submembrane plaque assembly, membrane distortion and internalization and recycling of membrane components are all important in maintaining normal cellular physiology [3,18,25,30,36,44,45,46,49,60]. Therefore, the early addition of membrane-associated cytoskeletal interactions to various versions of the Fluid–Mosaic Membrane model was considered very important [17,18]. The distributions and mobilities of integral membrane components can be modified or selectively anchored by cytoplasmic membrane-associated cytoskeletal components or by extracellular interactions (cell–cell, cell–matrix or cell–stromal interactions), resulting in cell membrane domain immobilization [56,58,59,60]. Such interactions also contribute to cell polarity and tissue organization [3,56,60]. 

**Biomembrane General Principle No. 13:** 
*Biomembranes possess specialized domain structures for extracellular and intracellular signaling and communication.*


One of the more important concepts in describing biomembranes over the years has been describing the linkages between the structural, organizational and dynamic aspects of membranes and the functional properties of cells, such as the communication of specific signals. For example, cell signaling and inter-cell communication are essential for maintaining various tissues and circulating cells. Cellular communications can take many different forms, for example, ions and transmitters in nerve transmission, hormones and other mediators, immune communication signals and many other examples, and these often involve specialized membrane domains. Often, individual communications and signals are transmitted into the cytoplasmic compartments of cells via dynamically assembled cell surface receptors and membrane-associated complexes in domains that dynamically include cytoskeleton systems [18,24,25,27,29,44,51,58,59]. As mentioned above, the cytoskeleton can also generate mechanical forces that can laterally move membrane complexes, membrane platforms, domains and even entire cells, or can inhibit their movements to help cells resist exterior mechanical forces [56,58,59,60,66]. The serial assembly of cytoplasmic proteins and cytoskeletal elements in and around membranes into specialized domains may be essential in the conversion of biochemical signals into mechanical forces that can influence cellular behavior, such as cell movements and organization of tissue structure [25,28,56,58,59,60]. There are a variety of membrane peripheral proteins and enzymes that have been identified as components involved in membrane–cytoskeletal interactions [24,25,28,51,56,58,59,60]. Membrane lipids are also involved in membrane–cytoskeletal interactions, resulting in the formation of specialized lipid signaling domains usually known as lipid rafts [30,44,45,46,49,50,51]. Although cytoskeletal involvement in specialized lipid domains was unknown at the time that the original Fluid–Mosaic Membrane model was proposed, the formation of lipid domains in biomembranes was foreseen years before actual experimental evidence for their existence was obtained [17].

Lipid membrane domains appear to form spontaneously as dynamic structures that result from the specific sorting of bulk lipid components, with some specific lipid-binding proteins or glycoproteins in structures held together mainly by noncovalent bonds [43,44,45,46,47,49,50,51]. It is not known if ligand or ion binding plays a role in lipid domain or lipid raft formation, but these events are likely to occur after these structures have spontaneously formed the membrane. The involvement of cytoskeletal transmembrane interactions with an assembled lipid raft or domain is likely a secondary event for initiation of transmembrane signaling [46,50,51]. The initial part of this process appears to be the presence of glycosylphosphatidylinositol (GPI) anchors at the cell surface in lipid domains or rafts [46,47]. The covalent tethering of specific GPI-bound proteins to specific phospholipids may be the first event in the formation of a lipid-domain signaling platform, or this event may occur after the domain has formed [44,45,46,47]. 

Depending on the cell type and cellular activity, GPI-anchored proteins in cell membranes may exist in different configurations or in unique domains, and in the process, this could result in many different, specific signaling platforms. For example, GPI-anchored proteins in lipid domains or rafts could be involved in cell signaling, cell adhesion or in other cellular processes [44,45,46,49]. Individual GPI-anchored proteins may also exist in different modes of lateral dynamics, perhaps without any detectable movement, or with completely free movement or free diffusion. Or these specialized domains might be similar to some cell surface receptors and show anomalous diffusion or transiently confined diffusion [27,31]. At the cytoplasmic membrane surface, some GPI lipid domains appear to be dynamically capable of being linked to cortical actin-containing cytoskeletal structures. This could explain some of the diffusion patterns seen with GPI-anchored proteins and their spatiotemporal properties [50,51]. 

Cell adhesion, spreading and motility are cellular processes that may be governed by the formation of tiny nanoclusters of small domains, some containing GPI-anchored proteins [50,51]. For example, Mouritsen described a membrane receptor signaling pathway that requires the formation of GPI-anchored protein nanoclusters [54]. This signaling pathway (RhoA signaling) is initiated by the binding of extracellular proteins containing the Arg-Gly-Asp binding motif, which can attach to cell surface β1-integrins. Binding to β1-integrin receptors eventually activates src focal adhesion kinases at the inner membrane surface, initiating the development of a cascade that includes actin nucleation by specific molecules (formins) and then actin–myosin contraction, all resulting from transmembrane linking and nanoclustering of membrane proteins. The result is a mechano-signaling process involving direct coupling of cellular actomyosin machinery to inner cell membrane lipids to functional GPI-anchored protein ligand-binding nanoclusters at the outer cell membrane surface [67].

There are likely many membrane domains on outer cell membrane surfaces that can activate specific peripheral and membrane-associated proteins at the inner cell membrane surface to form transmembrane domains, platforms or plaques in order to initiate cellular signaling [31,54]. This process starts with ligand-binding, membrane reorganization, immobilization of membrane domains, transmembrane signaling and activation of cytoplasmic enzymes and mechanocontractile processes, and it can be used for many cell activities. For example, it can also be used to signal internalization of plasma membrane domains in endosomes [5,68]. These completely integrated mechano-structures exist within single cells, groups of cells and tissues [3,5,68]. 

Extracellular signals from the microenvironment are constantly bombarding cells, and these cells must have filtering mechanisms to sort out this information and pass on the important and relevant signals to the cell’s interior. Specialized receptor structures at the cell surface are the first level of filtering extracellular signals, followed by the need for dynamic changes and assembly of complex signaling structures to provide additional filtering and facilitate the transmission of signals [69]. Cell membranes, at their inner surfaces, are also constantly interacting with and transmembrane linking to various structural components and enzymes in order to filter, process and amplify signals from the microenvironment, and they then pass these signals on to stimulate appropriate cellular responses. Cell membranes are also capable of sending messages back into the extracellular environment by releasing signaling molecules and membrane-encapsulated structures or by providing appropriate molecular signaling patterns to adjacent cells [5].

## 5. The Different Distributions and Lateral Mobilities of Biomembrane Components

Cell membrane components appear to display unique rotational and lateral mobilities and distributions due to their individual properties and a variety of restrictions on their rotational and lateral movements. These restrictions can also affect their residence times in various compartments and their confinements within assorted membrane domains [3,23,24,25,31,37,67]. These different lateral mobilities appear to depend on differences in local membrane compositions, spatial organizations, linkages and obstacles that are different from the average cell-membrane microenvironment [3,18,23,25,27,29,30,31,32,37,38,39,40,41,47,48,49,50,51,56,58,59,60]. For example, the lateral movement of some integral cell membrane proteins in the membrane plane can be restricted by multiple cis- and transmembrane interactions that constrain or direct their movement within or between various membrane domains. These modulators of distribution and movement occur within membranes, but can also include: extracellular interactions, such as binding to extracellular matrix and stroma; and intracellular interactions with peripheral and membrane-associated cytoskeletal structures [3,18,24,25,27,31,51,54,56,57,58,59,60,64,65,66,67]. 

**Biomembrane General Principle No. 14:** 
*Biomembrane components and membrane domains display a wide spectrum of lateral movements and distributions that appear to be functionally important and related to interactions with specific membrane domains, structures and barriers.*


Biomembranes have become much more complex as data have accumulated over the last five decades, especially on the dynamic aspects of membrane structure and function. For example, researchers have recently discovered the presence of structural membrane barriers that interfere with the free diffusion of membrane components in the membrane plane. At the inner cell membrane surface, a variety of peripheral membrane barriers, such as curvature-causing peripheral membrane proteins, cytoskeletal components and other obstacles, can place limits on membrane component distributions and movements [3,18,20,21,22,23,24,25,27,31,56,57,58,59,60]. There are a variety of distinct interactions that can occur at the inner and outer cell membrane surfaces, such as with a number of membrane complexes, domains, barriers, platforms and membrane-associated structures [17,18,20,21,22,23,24,25,29,31,56,57,58,63,64].

The restraint on mobility of integral membrane glycoproteins in the cell membrane plane and their presence in specific membrane domains has functional consequences [3,24,25,27,31,36,44,50,54,56,57,58,59,60,66]. The lateral movements of a few membrane proteins or cell surface receptors have been examined, and their movements (or restraint of movements) have been organized into various categories: (*a*) random movement or free diffusion in the fluid portions of the membrane; (*b*) transient movements confined by membrane obstacles made up of protein clusters that have been likened to ‘fence posts’ or ‘pickets’; (*c*) transient movements that are constrained by structural domains or ‘corrals’ circumscribed by cytoskeletal elements and their attachment molecules; or (*d*) directed movements due to attachment to and contraction of the cytoskeleton [24,25,27]. This has been interpreted as various motions of membrane components as: (*a*) free Brownian diffusion; (*b*) anomalous diffusion caused by changes in lipid nano-environment; (*c*) channeled diffusion restrained by membrane-associated cytoskeletal structures; (*d*) confined diffusion restrained by defined structural ‘corrals’; and (*e*) hop diffusion between dissimilar domains [24,25,27,31]. For example, the original Singer–Nicolson description of integral membrane proteins freely diffusing in the membrane plane is relevant to one of these categories [8]. 

Contemporary concepts of cell membrane dynamics dictate that substantial portions of integral membrane proteins are in mosaic structures that are incapable of free lateral diffusion in the cell membrane plane. They may be transiently capable of undergoing free diffusion in the membrane plane, but they are not freely mobile [18,22,23,24,25,27,28,31,63]. Some cell membrane components are thought to be wholly or partially confined to membrane domains circumscribed by membrane barriers or barriers attached to the membrane surface [22,23,24,25,27,31,40,49,55,63,65]. Since cell membranes are dynamic structures, some integral proteins and lipids may escape from one domain and move to adjacent domains or escape membrane domains altogether. They can also associate in the membrane plane and become supersized mosaic structures [22,24,25,60,63]. Supersized membrane structures may also be important in internalizing or releasing endosomes or exosomes. The abilities of membrane lipids and proteins/glycoproteins to move between adjacent membrane domains may be limited by the extent of their aggregation with similar or different components, the sizes of membrane barriers to movements and the complex interactions of membrane barriers with cytoskeletal elements and extracellular matrix or stromal components [3,24,25,57,60].

With the realization that membrane domains are dynamic, functional structures—the sizes of domains, their interactions, dynamics and linkages to membrane-associated structures can vary quite dramatically, depending on a number of factors. For example, they can exist dynamically as small lipid domains or rafts or as larger, more complex glycoprotein–lipid–membrane-associated-cytoskeletal domains that can also be linkages to other structures. The estimated or approximate areas of various membrane domains can vary from 0.04 to 0.24 μm^2^ (these have been described as micro- and submicrometer-sized domains). The approximate domain transit times of some membrane glycoprotein receptors can range from 3 to 30 s. Smaller membrane domains, such as nano- or meso-sized domains (of diameter 2–300 nm) are also present. In addition, barriers to the motion of integral membrane proteins are present. These complex actin-containing cytoskeletal fence domains can vary in diameter from 40–300 nm. This can be compared to small lipid raft domains that are usually in the diameter range of 2–20 nm [24,25]. Dynamic, integral membrane protein-complex domains are also present and can vary in size, with a minimum range of 3–10 nm in diameter (containing only a few components) to a maximum size of at least one hundred times this diameter [24,25]. Most cells have several different types of cell membrane domains, and evidence suggests that some of these domains are present as cell surface signaling complexes. This indicates that there is another, higher level of membrane organization and complexity beyond the original description of the Fluid-Mosaic Membrane Model [3,8,18]. Kusumi et al. [24,25] called this more-complex representation Hierarchical Membrane Organization.

The hierarchical organization of membrane structures is based on several different observations of cell surface receptor dynamics. For example, the variability and dissimilarity of lateral motions of various cell surface receptors and other membrane components as well as the ability of cells to quickly change their cell surface membranes in order to respond to intracellular and extracellular signals supports a hierarchical organization [24,25]. Thus biomembrane organization may have evolved so that cells can rapidly and selectively respond to numerous specific extracellular signals. It may be more efficient to have various receptors prepositioned on the cell surface within untriggered signaling domains that can be rapidly and specifically capable of aggregating into supramolecular transmembrane signaling structures [25]. The presence of membrane protein barriers or ‘fences’ on the inner plasma membrane surface can limit the range of lateral motion of integral membrane protein components. Some examples include limiting lateral motion within cytoskeletal-fenced ‘corrals’, or tethering them directly or indirectly to membrane domains. This may create more stable, local membrane domains of high receptor densities that do not have to be nascently synthesized. Such hierarchical structures can incorporate membrane domains with cell surface receptor diffusion rates that are 5- to 50-times lower than the same components in freely diffusing membrane environments. Therefore, some receptors can be confined on the average to specific membrane subregions with restricted mobilities and ranges of display [24,25]. The prepositioning of receptors in more-dense arrays so that they are more capable of ligand binding without requiring extensive lateral rearrangements should increase the efficiency of response to an extracellular signal.

We can now propose that the prerequisites of some (and probably many) cell signaling systems that involve cell membrane receptor–ligand binding are basic Fluid–Mosaic membrane structures and specific membrane domains capable of forming ligand–receptor clusters surrounded by fluid-phase lipids. In addition, many signaling domains should be transmembrane-linked to membrane-associated signaling systems on the cytoplasmic side of the plasma membrane [18,24,25]. A membrane signaling compartment or signaling domain can be further defined by whether aggregations of similar or different domains are required, or their confinement to signaling ‘zones’ by cytoskeletal or protein fencing at the inner surface, in addition to other enzymatic properties, are important in the overall signaling process [24,25]. 

Membrane receptors and their display as different arrangements in membrane domains and other structures likely represents a normal situation in cell membranes in order to accommodate the large numbers of possible extracellular signals that cells receive. In this way, particular signals can be distinguished from one another. An example of the possible display of hypothetical receptors in a Fluid-Mosaic membrane containing multiple lipid domains, glycoprotein complex domains, barrier or ‘corral’ domains and other membrane-associated structures is portrayed in Figure 2 [3,18]. Such schemes should not be taken too seriously, because they will likely undergo further changes as soon as new data are available. Cell membrane structures must also be dynamic and affected by a variety of conditions, such as the binding of various extracellular molecules, changes in intra- and extracellular ion and solute concentrations and integration of lipid molecules from inside and outside the cell. 

Cells appear to utilize different types of cell membrane domains to manage cellular physiology. In addition to the dynamic membrane domains involved in cell signaling, nutrient transport and other properties, cells also have less dynamic, more stable mosaic membrane structures that are involved in preserving cell polarity, stable cell–cell interactions and tissue organization. These latter properties may require more mosaic (more structured) and less mobile receptors that are more integrated and linked to intracellular cytoskeleton structures as well as extracellular structures in pericellular spaces. The extracellular and junctional structures found between cells in tissues are also transmembrane-linked to peripheral membrane proteins and membrane-associated cytoskeletal elements to form integrated tissue networks. Such networks play an important role in the tensile forces and mechanical viscoelastic responses of cells in tissues [70,71,72]. 

**Biomembrane General Principle No. 15:** 
*Biomembranes undergo dynamic changes in domain mobility, size, area and structure with assembly and disassembly of various components reacting to changes in the microenvironment and the receiving and sending of cellular communication signals.*


This general principle of biomembrane models highlights some of the organizational and dynamic aspects of biomembranes that are difficult to present in static diagrams and figures. The greatest differences in the newer, ever-evolving Fluid–Mosaic Membrane Model are the additions of more membrane-associated elements and the enhanced closeness of molecular relationships or higher densities (mosaic nature) of cell membrane components. Figure 2 depicts a rather simplified schematic of these additions to the Fluid–Mosaic Model. As stated previously [3,18], we cannot take such schemes too seriously, because they shall surely change again over time as more information is revealed about the structure and dynamics of cell membranes.

## 6. Movements of Lipids Into/Out of Cells: Lipid Carriers, Vesicles, Globules and Droplets

Once some of the general principles of cell membrane structure, organization and dynamics have been established, we can ask whether these are important in the turnover or trafficking of membrane components, such as membrane lipids. The health of tissues and cells is highly dependent on their ability to capture and transfer nutrients and remove cellular waste, such as toxic or damaged molecules. Once nutrients have been transported into cells and are present in intracellular spaces, such as between various organelles and cellular compartments, they must be delivered to various intracellular membranes and organelles. Consequently, the abilities of cells to rapidly move various nutrients, structural components and newly synthesized molecules to where they are needed intracellularly and to remove them if they are damaged and no longer needed are essential to cell, tissue and organ health. This normal trafficking of cellular molecules is especially important and crucial for membrane lipids [73,74,75,76,77,78]. 

The transport of lipid molecules into cells (or their intracellular biosynthesis) and their eventual delivery to various cell organelles and intracellular membranes (or their secretion to the extracellular microenvironment) generally requires their movement by specific lipid transport molecules or their incorporation into small membrane vesicles, lipid globules or other delivery systems. These transport systems move lipids to specific intracellular membrane sites (or the plasma membrane) or to domains at specific membrane or domain sites [76,79,80]. Alternatively, different intracellular membranes can be used to deliver membrane lipids to distinct membrane sites, resulting in transient fusion and exchange of membrane constituents with other membrane compartments and organelles [79,80]. Such processes can be used to repair damage to the plasma membrane and intracellular membranes by removing damaged molecules and replacing them with undamaged molecules in order to maintain cell function [81,82]. 

**Biomembrane General Principle No. 16:** 
*Cells use a variety of biomembrane and related structures, such as intracellular membranes, lipoproteins, membrane vesicles,*
*chylomicrons and lipid globules and droplets, to move various membrane lipids into and out of various cell compartments and to remove damaged membrane lipids from organelles and cells.*


Here we expand briefly on some of the relationships between dietary membrane lipids and various cellular membranes, focusing on the movements of membrane lipids between different membranes and the removal of damaged membrane lipids from cells. First, it has been established that dietary lipid sources can be used to drive the replacement of damaged membrane lipids, even though it is usually impossible to consume sufficient quantities of membrane lipids in foods to fulfill this replacement by diet alone [78]. Dietary lipids, including membrane lipids, are ingested, digested, absorbed by epithelial cells in the upper small intestines and then transferred and transported via lymph and blood circulation to the liver and to various organs and tissues to be absorbed again. Eventually, membrane lipids are moved around within cells using carriers, such as lipoproteins, lipid-binding proteins, chylomicrons, small lipid globules, membrane vesicles and intracellular membranes (Figure 3) [73,74,75,76]. Inside cells, the carrier membrane vesicles, intracellular membranes, chylomicrons and various lipid globules and droplets fuse with target membranes to deliver membrane lipids and to replace and remove damaged membrane lipids in a reverse process. This entire in vivo delivery and exchange process works on a concentration gradient system by the principle of mass action or bulk flow [81]. The lipid-to-membrane and membrane-to-membrane fusion events occur continuously in cells, and these are crucial in delivering or redistributing membrane lipids between various intracellular membranes and between different cellular compartments. 

The movements of lipids within cells via various lipid transport systems are quite dependent on lipid properties. These properties include: the structure, composition, distribution and acylation of lipids in the transported lipid as well as the transport vehicle and the target membrane domains where membrane binding and fusion occur. Other specialized components are also required for successful lipid delivery. In addition to the presence of fusogenic proteins, specific electrolytes and other essential molecules, the properties of membranes at the point of fusion are also crucial [84,85,86,87]. In some cases, lipid compositional changes can be used to track the transfer of specific lipids. For example, the delivery of sphingolipids to specific cellular membranes can be traced by analyzing the presence of specific sphingolipids in target membranes [48,88]. Sphingolipids are commonly found to be concentrated in intracellular vesicles destined to fuse with the plasma membrane, and thus the presence of specific membrane lipids appears to be important in the specific targeting of specialized lipids to specific membrane domains [48,81,87]. 

Membrane fusion is an essential part of the process of lipid delivery to specific membrane sites [84,85]. Membrane fusion is a rather common biological event, and it occurs in a number of normal events, such as fertilization, development, endocytosis, secretion, nerve transmission and many other normal developmental and restorative processes. Membrane fusion is also important in many pathologic conditions, such as infection, inflammation, neoplasia, cell death and other events [84,85,86,87,88,89]. 

An important restraint on membrane fusion is the necessity for specific ‘fusogenic’ proteins. Such proteins are necessary to temporarily bind adjacent membranes together long enough for membrane fusion to take place [84,85,87,89]. This process requires the close apposition of the two membranes, along with the presence of the fusogenic proteins and the countering of repulsive electrostatic forces between the fusing membranes. After close apposition, destabilization of the bilayer lipid structures occurs so that the lipids can form into a temporary non-bilayer transition structure, followed by the rapid reunification of the membrane lipids into bilayer structures [84,85,86,87,89]. 

In secretory cells, the presence of dedicated fusogenic machinery at the plasma membrane inner surface is notable. One example of this machinery is a specialized structure made up of a dynamic membrane microdomain specialized for secretion, called the ‘porosome’ [88,90]. As visualized in electron micrographs, porosomes appear to be ‘pits’ approximately 0.5–2 μm in diameter with depressions of 100–180 nm [91]. Porosomes are required for some normal cell secretory functions, such as the secretion of proteins, glycoproteins, enzymes, bioregulators and other important molecules [90,92]. 

There are other less specific membrane structures that can release lipids and other molecules from cells. In addition to specialized plasma membrane secretory systems, cells also spontaneously bleb and release various plasma membrane vesicles [5,93,94]. Some of these released membrane vesicles, or exosomes, are specialized for delivery of non-membrane molecules. Exosomes are a particular class of released membrane vesicles that also have incapsulated soluble proteins and nucleic acids [95,96,97]. Thus, exosomes represent specialized membrane vesicles with potential cell-to-cell communication properties, and some of these may be important in the regulation of normal physiological processes [95,96]. Thus, released membrane vesicles may also be involved in the trafficking of membrane components and the removal of damaged plasma membrane molecules from cells and tissues [5,97].

## 7. Membrane Lipid Replacement with Dietary Membrane Phospholipids

As briefly discussed in the sections above, unsaturated glycerolphospholipids and other lipids in biomembranes are eventually damaged (mainly by oxidative reactions), degraded or destroyed and must be repaired or replaced to maintain normal cellular membrane function and cellular physiology [3,83,98,99]. The polyunsaturated fatty acids in cellular membranes are particularly susceptible to free radical oxidative damage, and this type of damage occurs universally during aging and disease [98,99,100]. Accordingly, dietary replacement of damaged membrane phospholipids with undamaged, functional phospholipids is essential in maintaining cellular and organ function and general health [33,101,102,103]. However, maintaining fully functional cellular membranes with only a dietary source for replacement of membrane phospholipids is often quite difficult, especially in patients with chronic or acute illnesses. Therefore, dietary supplements have been added to diets for augmenting the intake of membrane lipids (Membrane Lipid Replacement (MLR)) [33,102,103]. 

Since membrane phospholipids can be oxidized, degraded or enzymatically modified before ingestion and prior to absorption within the gastrointestinal system, the protection of membrane phospholipids before and during bioabsorption is essential for successful MLR [104,105]. Such protection can be achieved by the addition of specific fructooligosaccharides (or inulins) to the membrane phospholipids, which directly insert into glycerolphospholipid bilayers between adjacent head groups and protect them from oxidation, excess temperatures, acidity, phospholipases and bile salts [106,107]. For MLR lipid supplements to be effective, the lipids must be protected from oxidation and degradation [33,102,103]. Because of the mass action, bulk flow principles that govern membrane lipid transport and MLR, the purity and protection of membrane phospholipids are crucial.

When MLR phospholipids are present in excess concentrations in the gastrointestinal system, most of these MLR phospholipids are promptly absorbed by the small intestines as undegraded small lipid globules and micelles, not as individual molecules transported separately. Bioabsorption of individual membrane lipid molecules is a much less efficient process that utilizes membrane carrier or transfer proteins. Irrespective of the actual mechanism of gut epithelial bioabsorption, the overall collective process is very efficient, and over 90% of ingested membrane phospholipids are absorbed and transported from the gastrointestinal system into the blood within a few hours [104,105]. While in the blood circulation, membrane phospholipids are usually protected by their association with carrier systems, such as lipoproteins or blood cell membranes. During MLR supplementation when membrane phospholipids are present in vast excess, they are mostly absorbed as small phospholipid globules and micelles. Eventually the MLR phospholipids are transported to tissues and cells, where they are transferred into cells by direct membrane contact, endocytosis or by specific carrier and transport proteins. Once inside cells, membrane phospholipids can be moved to various cellular compartments and organelles by a number of mechanisms, including membrane–membrane transfer, carrier molecules, small lipid globules, membrane vesicles, chylomicrons and other mechanisms [33,103,105]. During this transfer process and after their intracellular delivery, the membrane lipids can be enzymatically modified by head group exchange or by enzymatic changes in fatty acid side chains and saturation to reflect the specific and everchanging needs of the membranes at their final destinations [33,102,103]. 

As mentioned repeatedly above, the entire process of membrane lipid uptake, transport, replacement, exchange and removal is driven overall by a mass action or bulk flow mechanism [82]. Thus, when protected membrane phospholipids are in vastly excess concentrations during MLR, they have the advantage of being able to reach their final intracellular destinations more efficiently and with significantly less degradation or free radical oxidation than unprotected dietary lipids. The mass action basis of bulk membrane lipid uptake, transport and delivery to intracellular membranes is also true of the reverse of this process, which eventually results in the exchange and removal of damaged, oxidized phospholipids and their transport to and elimination via the gastrointestinal system [74,75,77,82]. 

### 7.1. Membrane Lipid Replacement in Chronic Illnesses and Environmental Exposures

MLR, the use of oral dietary supplements containing protected, essential polyunsaturated glycerolphospholipids and other membrane lipids, has been successfully used to maintain and recover lost or reduced mitochondrial and membrane function [33,102,103]. The most common dietary therapeutic use of oral MLR phospholipids is to reduce fatigue and improve mitochondrial function [33,102,103]. Fatigue is the most common complaint of patients seeking general medical care, and excess fatigue is associated with aging and most if not all chronic and many acute medical conditions [108]. Fatigue is not well understood at the cellular level; it can be perceived as a loss of overall energy, extreme mental and/or physical tiredness, exhaustion and/or diminished endurance. It can also be apparent as a loss of function, combined with an inability to perform even simple tasks without exertion [108,109]. With age and in chronic and most acute diseases, fatigue is regularly present due to a variety of causes. For example, in individuals with complaints of moderate to severe fatigue, their fatigue has been directly related to loss of mitochondrial function and diminished production of ATP by mitochondria [109]. 

Fatigue of long-term duration has been termed chronic fatigue, and this has been seen in a variety of chronic illnesses, especially chronic fatiguing illnesses. An example of this is chronic fatigue syndrome or myalgic encephalomyelitis (CFS/ME). CFS/ME is a condition that features fatigue of long-term duration. Although fatigue is usually the primary complaint, CSF/ME is a condition that involves multiple complaints and signs and symptoms [110,111,112]. Moreover, in almost all chronic illnesses, fatigue is a common secondary complaint [108,109,112]. Although severe fatigue is uniformly related to significant loss of mitochondrial function and production of ATP, mild fatigue can be found in depression and in some other psychiatric conditions [108,109]. 

MLR has been used for reducing fatigue in patients with chronic fatigue and other chronic and fatiguing illnesses [33,102,103,112]. For example, in middle-aged to older subjects (61–77 years-old) in a crossover clinical study, the MLR supplement NTFactor Lipids^®^ was used to treat chronic fatigue symptoms while mitochondrial function was monitored. There was good correspondence between reductions in fatigue scores and improvements in mitochondrial function tests [113]. These subjects with moderate to severe chronic fatigue showed significant improvements in fatigue and other clinical parameters during the test arm of the study, but the improvements were slowly reversed when patients were switched to the placebo arm of the study. The improvements in mitochondrial function assessed by inner mitochondrial membrane transmembrane potential matched the clinical data and showed enhancement up to 45% while on the MLR supplement, but these gains were slowly reversed after the patients were switched to the placebo arm of the study [113]. Similar positive results on the effects of MLR phospholipid supplements on reducing fatigue from 26–43% were found in various chronic conditions, including CFS/ME, fibromyalgia, Gulf War illness, chronic Lyme disease and other infections, and various cancers [33,102,103,112,113,114,115].

Recently, we examined the ability of MLR supplements to reduce the severities of several signs and symptoms in environmentally exposed patients [116,117]. Case reports indicated that chemically exposed veterans with multiple-symptom conditions benefited significantly from MLR with NTFactor Lipids^®^ [116], so a study was initiated to examine the effects of 6 g per day of NTFactor Lipids^®^ on multiple signs and symptoms during 6 months of continuous MLR with oral glycerolphospholipids. The severities of over 100 signs and symptoms were reported at various times using patient illness survey forms, and the clinical data were combined into various symptom categories [117]. This clinical study showed that there were gradual and significant reductions of symptom severity in categories related to fatigue, pain, musculoskeletal, nasopharyngeal, breathing, vision, sleep, balance, urinary, gastrointestinal and chemical sensitivities after 6 months of MLR with 6 g per day of membrane glycerolphospholipids (Figure 4). This preliminary study indicated the potential for using MLR with membrane phospholipids to improve health outcomes in patients with environmental exposures [117]. Although the mechanism(s) of symptom reductions were not determined in these studies, the reductions in symptoms might eventually be related to a combination of slow chemical removal from deeply embedded tissue stores and enhancement of mitochondria function.

### 7.2. Membrane Lipid Replacement and Chronic Pain

One of the symptom categories in the studies discussed in the previous section was pain, and MLR supplements, such as NTFactor Lipids^®^, have been used to help reduce widespread musculoskeletal pain, peripheral neuropathy and gastrointestinal symptoms, like stomach pain, in chronically ill patients [115,116,117,118,119]. Pain is a complex phenomenon that can be initiated by injury, illness or environmental exposures. Pain is usually categorized by varying criteria based on its pathophysiological mechanism, duration, etiology and anatomical source [120,121]. One type of pain that is often widespread is nociceptive pain. This has been described as acute or chronic pain, or as a sharp or throbbing pain that is experienced in the joints, muscles, skin, tendons and bones. Nociceptive pain is usually considered a short-lived condition, although it can also be chronic. This type of pain is often generated in response to potentially harmful stimuli, and it can be divided into two categories: somatic nociceptive pain, which is usually localized in the dermis, and visceral nociceptive pain, which usually arises as diffuse and poorly defined pain sensations in the midline of the body. Either type of pain can be caused by multiple events acting on nociceptors to induce pain sensations [120,121]. The nerve membrane channels that are involved in nociceptive pain have been identified as Transient Receptor Potential (TRP) channels (TRPV1, TRPM3, TRPA1, etc.) [122]. This family (more than 50 subtypes) of membrane channels has become a popular therapeutic target for the development of new treatments for chronic pain [123]. 

The TRP channel superfamily in mammals consists of 6 subfamilies and 28 members that mainly act as cation channels. These channels possess a primary structure that is common to all of its members, and this primary structure is comprised of 6 transmembrane domains and one hydrophilic loop that form a pore structure that is primarily permeable to monovalent cations, but in certain cases also calcium ions [124]. Some TRP channels are essential for nociception and thermal sensitivity [125,126]. 

Membrane lipids are important in the function of TRP channels. For example, some membrane channels require the existence of phospholipids for activity [125,126,127]. TRP channels are apparently regulated by membrane phosphoinositides, such as phosphatidylinositol 4,5 bisphosphate or PI(4,5)P_2_. Although this phospholipid was initially described as a general inhibitor of TRP channels, it can act as an agonist with desensitization properties. A fraction of phosphoinositol PI(4,5)P_2_ is present in NTFactor Lipids^®^, and it can inhibit the heat- and capsaicin-activation of TRPV1 channels. The breakdown of PI(4,5)P_2_ by phospholipase C alleviates this inhibition. This results in potentiation of TRPV1 activity by proinflammatory agents such as bradykinin [128]. Even if the TRP channels are activated by PI(4,5)P_2_, they quickly become unresponsive as they become desensitized, removing the ability to be stimulated [129]. 

Phospholipase C can catalyze the hydrolysis of PI(4,5)P2, resulting in the formation of the two classical second messengers (inositol 1,4,5 trisphosphate (IP3) and diacylglycerol (DAG)). A possible explanation for the effect of PI(4,5)P2 on TRP channels is that the negatively charged headgroup of PI(4,5)P_2_ interacts with positively charged residues in the cytoplasmic domains of TRP channels [130]. This was confirmed when the co-crystal structures of TRP channels with and without PI(4,5)P_2_ were published by Hansen et al. [131]. Although the specific mechanism of action of phosphoinositosides on TRP nociceptor channels is not fully understood, both of the proposed mechanisms (inhibition or activation with desensitization) result in a final decrease in the activity of these channels, either by inhibition or desensitization. The final result is a decrease in pain sensitivity and nociception promoted by glycerolphospholipids, which are present in MLR supplements such as NTFactor Lipids^®^ [102,103].

The requirement of higher doses, for example 6 g per day of NTFactor Lipids^®^ [116,117], to significantly inhibit pain may be justified by the fact that PI and its derivatives are minor phospholipids in MLR supplements [102,103]. Interestingly, most patients on MLR supplements, such as oral NTFactor Lipids^®^, gradually moved to higher daily doses of the supplement to control pain [116]. There are other possible explanations for the inhibition of pain by MLR supplements, such as stabilization of nerve membrane resting potential and inhibition of depolarization. These are now under examination by the authors to determine the role of MLR phospholipids in pain reduction.

## 8. Final Comment

Models of cellular membranes have evolved to be considerably more complex as well as more compact or mosaic than the diagrams presented in the original Singer–Nicolson Fluid–Mosaic Membrane Model [8]. Although newly published information on membrane structure, organization and dynamics, briefly presented here in an overview, has been generally accepted by the scientific community, we are just beginning to understand the role of various cellular membranes and their domain properties in explaining complex biological phenomenon. This information will be essential in elucidating the complex interrelationships between cells in tissues and cells in fluid environments. It will also be indispensable in the development of new therapeutic approaches, such as MLR, that can overcome, at least in part, various pathological conditions that are linked to the loss of cellular membrane integrity, organization, dynamics and function.

## Figures and Tables

**Figure 1 biomedicines-10-01711-f001:**
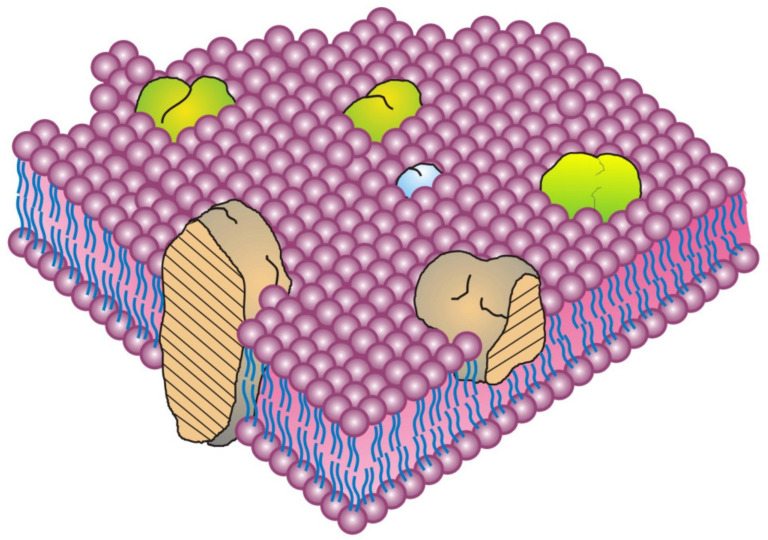
The Singer–Nicolson Fluid–Mosaic Membrane Model of cell membrane structure as proposed in 1972. In this static view of a fluid cell membrane, the solid bodies with stippled cut surfaces represent globular integral membrane proteins randomly distributed in the plane of the membrane. Some integral membrane proteins form specific integral protein complexes, as shown in the figure. Integral proteins are represented as intercalated into a fluid lipid bilayer, but peripheral membrane proteins are mentioned but not shown, nor are tightly-bound lipids. Further, the figure does not contain other membrane-associated structures or membrane domains (Redrawn from Singer and Nicolson [8]).

**Figure 2 biomedicines-10-01711-f002:**
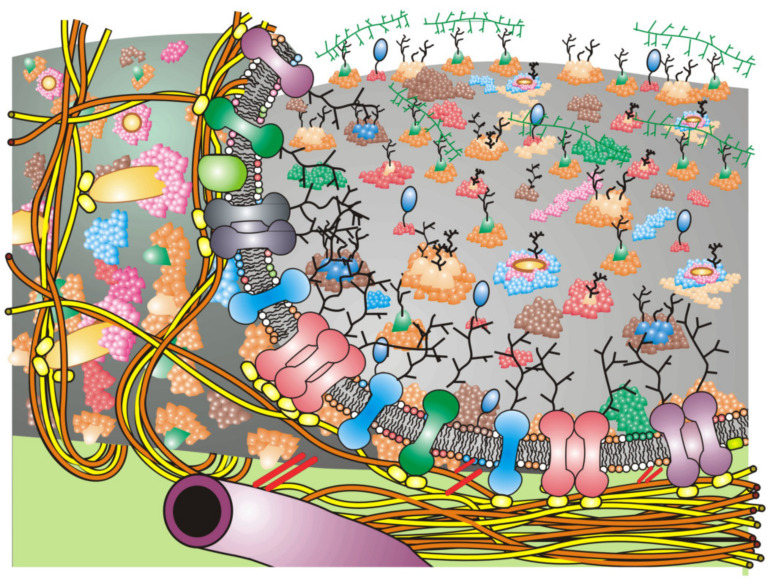
An updated, static representation of the Fluid–Mosaic Membrane Model with more detail than presented in the original 1972 Singer and Nicolson model [8]. The cell membrane of a generic tissue cell is depicted with various lipid and protein domain structures as well as membrane-associated cytoskeletal and extracellular structures. The cell membrane has been peeled back at the left in order to reveal underneath the plasma membrane. Membrane-associated cytoskeletal elements can be arranged to form potential barriers (‘corrals’) that could possibly limit the lateral mobilities of some of the integral transmembrane proteins. In addition, membrane-associated cytoskeletal structures can indirectly interact with some of the integral membrane proteins at the inner membrane surface along with stromal or extracellular matrix components at the outer surface. Although this static diagram presents some of the possible mechanisms of integral membrane protein mobility restraint, it does not accurately represent the dynamic changes in membrane components, the sizes and structures of phospholipids and lipid domains, integral and peripheral membrane proteins or membrane-associated cytoskeletal structures. It also does not reflect the actual crowding or high density of membrane components. (Modified from Nicolson [18]).

**Figure 3 biomedicines-10-01711-f003:**
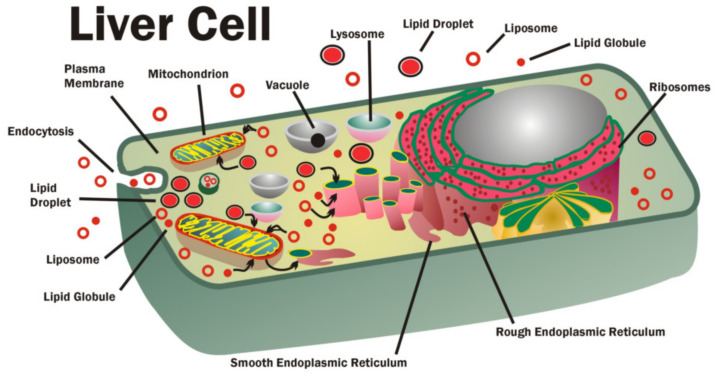
A few of the phospholipid transport structures involved in the delivery of membrane phospholipids and other lipids to intracellular membranes, and the reverse of this process to remove damaged lipids. A liver cell is shown with internal lipid transfer and storage systems, such as lipid micelles, globules, vesicles, chylomicrons and lipid droplets. These various lipid transport and transfer structures can bind to different intracellular membranes and transfer glycerolphospholipids and other lipids, and can pick up damaged lipids for eventual delivery to the extracellular environment. Not shown in the figure are lipid transport/transfer by direct adjacent membrane-to-membrane contact and lipid droplet-, globule-, chylomicron- and vesicle-to-membrane contact by temporary fusion with adjacent intracellular membranes. Both the forward and reverse processes appear to be driven by mass action or bulk flow mechanisms. (Modified from Nicolson et al. [83]).

**Figure 4 biomedicines-10-01711-f004:**
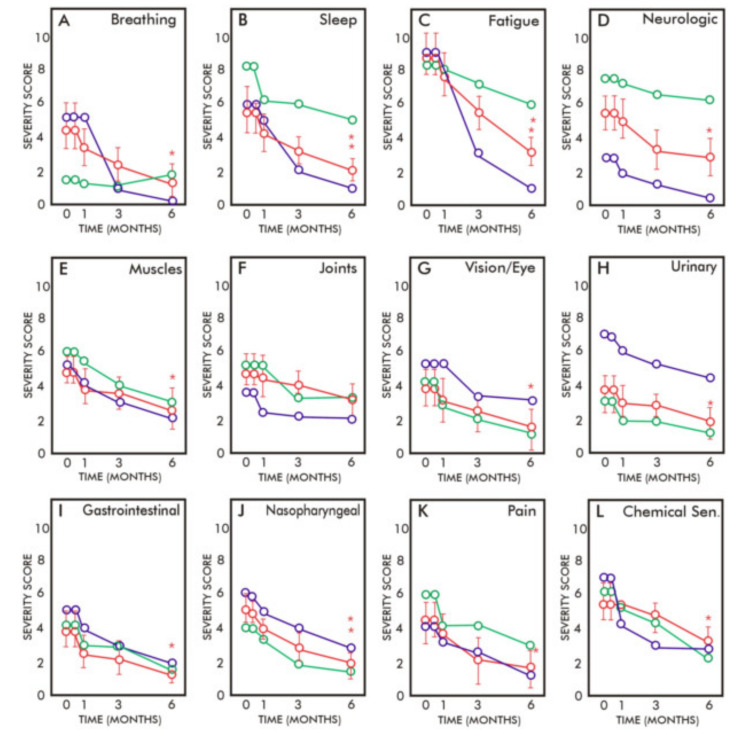
Symptom category scores with time reported by chemically exposed veterans who took the oral MLR supplement NTFactor Lipids^®^ (6 g per day) for 6 months. The mean symptom category severity scores of all 16 trial participants (red symbols with standard error of the mean) are compared to two individual subjects (green and blue symbols) before the trial and at one week, one month, 3 months and 6 months. Each symptom category represents the mean of 3–6 individual symptoms. (**A**) Breathing difficulties, (**B**) Sleep disturbances, (**C**) Fatigue, (**D**) Neurologic symptoms, (**E**) Muscle symptoms, (**F**) Joint symptoms, (**G**) Vision/eye disturbances, (**H**) Urinary symptoms, (**I**) Gastrointestinal symptoms, (**J**) Nasopharyngeal symptoms, (**K**) Pain, and (**L**) Chemical sensitivities. *, *p* < 0.01; **, *p* < 0.001. Modified from Nicolson and Breeding [117].

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
