# Peer review of "Fifty Years of the Fluid–Mosaic Model of Biomembrane Structure and Organization and Its Importance in Biomedicine with Particular Emphasis on Membrane Lipid Replacement"

_biomedicines, 2022, doi:10.3390/biomedicines10071711_

Round 1
Reviewer 1 Report
This review is a historical tour de force of the effect of a membrane model proposed in 1971 on subsequent cell biology and medicine. The senior author has a lifetime of significant research findings since the original paper was published during his PhD studies. I find no fault with the paper other than a problem with lines 467 and 991.
Author Response
This review is a historical tour de force of the effect of a membrane model proposed in 1971 on subsequent cell biology and medicine. The senior author has a lifetime of significant research findings since the original paper was published during his PhD studies. I find no fault with the paper other than a problem with lines 467 and 991.
Response: We thank the reviewer for his/her kind comments. We have changed the wording around line 991 (now line 1006) to read: Models of cellular membranes have evolved to be considerably more complex as well as more compact or mosaic than the diagrams presented in the original Singer-Nicolson Fluid–Mosaic Membrane Model [8]. The wording around line 467 (now line 485) was not changed. The authors did not feel that there was any problem with this statement as a principle of biomembrane structure. If the reviewer can be more specific with his/her problem with this statement of principle, we could address this problem by revising the statement.
Reviewer 2 Report
An excellent and timely review.
Author Response
An excellent and timely review.
Response: We thank the reviewer for his/her generous comment.
Reviewer 3 Report
This is a review about the history and relevance of the fluid–mosaic model of biomembrane structure and organization, with special emphasis on membrane lipid replacement. One of the authors has introduced this conceptual model 50 years ago and has since remained an eminent force in biomembrane research. At the core of the review are 16 general biomembrane principles, that the authors discuss and rationalize. This is an excellent perspective on the fluid–mosaic model that bridges its history with current understanding. In my view, it rightly argues that what we know now about biomembranes is much richer and more differentiated than 50 years ago, but the basic conceptual framework that the fluid–mosaic model offers remains valid. The model has grown over time without the need to alter its identity. I enjoyed reading the manuscript, and I recommend its publication without hesitation.
Author Response
This is a review about the history and relevance of the fluid–mosaic model of biomembrane structure and organization, with special emphasis on membrane lipid replacement. One of the authors has introduced this conceptual model 50 years ago and has since remained an eminent force in biomembrane research. At the core of the review are 16 general biomembrane principles, that the authors discuss and rationalize. This is an excellent perspective on the fluid–mosaic model that bridges its history with current understanding. In my view, it rightly argues that what we know now about biomembranes is much richer and more differentiated than 50 years ago, but the basic conceptual framework that the fluid–mosaic model offers remains valid. The model has grown over time without the need to alter its identity. I enjoyed reading the manuscript, and I recommend its publication without hesitation.
Response: We thank the reviewer for his/her kind comments.